# Molecular Subtyping of Invasive Breast Cancer Using a PAM50-Based Multigene Expression Test-Comparison with Molecular-Like Subtyping by Tumor Grade/Immunohistochemistry and Influence on Oncologist’s Decision on Systemic Therapy in a Real-World Setting

**DOI:** 10.3390/ijms23158716

**Published:** 2022-08-05

**Authors:** Ramona Erber, Miriam Angeloni, Robert Stöhr, Michael P. Lux, Daniel Ulbrich-Gebauer, Enrico Pelz, Agnes Bankfalvi, Kurt W. Schmid, Robert F. H. Walter, Martina Vetter, Christoph Thomssen, Doris Mayr, Frederick Klauschen, Peter Sinn, Karl Sotlar, Katharina Stering, Albrecht Stenzinger, Marius Wunderle, Peter A. Fasching, Matthias W. Beckmann, Oliver Hoffmann, Rainer Kimmig, Nadia Harbeck, Rachel Wuerstlein, Fulvia Ferrazzi, Arndt Hartmann

**Affiliations:** 1Institute of Pathology, University Hospital Erlangen, Friedrich-Alexander-Universität Erlangen-Nürnberg (FAU), 91054 Erlangen, Germany; 2Comprehensive Cancer Center Erlangen-EMN (CCC ER-EMN), 91054 Erlangen, Germany; 3Klinik für Gynäkologie und Geburtshilfe, Frauenklinik St. Louise, Paderborn, St. Josefs-Krankenhaus, Salzkotten, Frauen Und Kinderklinik St. Louise, 33098 Paderborn, Germany; 4Institute of Pathology Viersen, 41747 Viersen, Germany; 5Institute of Pathology, University Hospital Essen, University of Duisburg-Essen, 45147 Essen, Germany; 6Ruhrlandklinik, West German Lung Center, University Hospital Essen, University of Duisburg-Essen, 45147 Essen, Germany; 7Department of Gynecology, Martin Luther University Halle-Wittenberg, 06120 Halle, Germany; 8Institute of Pathology, Ludwig-Maximilian University (LMU), 80337 München, Germany; 9Institute of Pathology, University Heidelberg, 69120 Heidelberg, Germany; 10Institute of Pathology, Paracelsus Medical University Salzburg, A-5020 Salzburg, Austria; 11Molekularpathologisches Zentrum, Pathologisches Institut, Universität Heidelberg, 69120 Heidelberg, Germany; 12Department of Obstetrics & Gynecology, University Hospital Erlangen, Friedrich-Alexander-Universität Erlangen-Nürnberg (FAU), 91054 Erlangen, Germany; 13Klinik für Frauenheilkunde und Geburtshilfe, Universität Duisburg-Essen, 45147 Essen, Germany; 14Breast Center and CCC Munich, LMU University Hospital, Department of Obstetrics and Gynecology, 80337 Munich, Germany; 15Department of Nephropathology, Institute of Pathology, Friedrich-Alexander-Universität Erlangen-Nürnberg (FAU), 91054 Erlangen, Germany

**Keywords:** Prosigna, PAM50, multigene expression analysis, immunohistochemistry, IHC, tumor grade, breast cancer, chemotherapy

## Abstract

In intermediate risk hormone receptor (HR) positive, HER2 negative breast cancer (BC), the decision regarding adjuvant chemotherapy might be facilitated by multigene expression tests. In all, 142 intermediate risk BCs were investigated using the PAM50-based multigene expression test Prosigna® in a prospective multicentric study. In 119/142 cases, Prosigna® molecular subtyping was compared with local and two central (C1 and C6) molecular-like subtypes relying on both immunohistochemistry (IHC; HRs, HER2, Ki-67) and IHC + tumor grade (IHC+G) subtyping. According to local IHC, 35.4% were Luminal A-like and 64.6% Luminal B-like subtypes (local IHC+G subtype: 31.9% Luminal A-like; 68.1% Luminal B-like). In contrast to local and C1 subtyping, C6 classified >2/3 of cases as Luminal A-like. Pairwise agreement between Prosigna® subtyping and molecular-like subtypes was fair to moderate depending on molecular-like subtyping method and center. The best agreement was observed between Prosigna® (53.8% Luminal A; 44.5% Luminal B) and C1 surrogate subtyping (Cohen’s kappa = 0.455). Adjuvant chemotherapy was suggested to 44.2% and 88.6% of Prosigna® Luminal A and Luminal B cases, respectively. Out of all Luminal A-like cases (locally IHC/IHC+G subtyping), adjuvant chemotherapy was recommended if Prosigna® testing classified as Prosigna® Luminal A at high / intermediate risk or upgraded to Prosigna® Luminal B.

## 1. Introduction

Invasive breast cancer (IBC) is the most common malignant tumor in women regarding morbidity and mortality. In 2016, the annual number of newly diagnosed breast cancer patients in Germany was 69,660 and 18,736 patients died due to the disease [1]. Therapy options include surgical and radiation therapy as well as systemic therapy such as endocrine therapy and chemotherapy. The treatment strategy is determined individually for each IBC patient, based on biology of both tumor and patient but also according to international and national guidelines [2,3,4,5,6]. Deciding whether a patient with IBC should receive systemic therapy depends on both prognostic and predictive factors. Prognostic (and partly predictive) factors in early, non-metastatic IBC that has not metastasized are: age, tumor size, nodal status, tumor grade, proliferation, angioinvasion, hormone receptor (HR) status, HER2 status [7], and molecular subtype [8]. In general, early HR positive, i.e., estrogen receptor (ER) and/or progesterone receptor (PR) positive, IBC frequently responds to endocrine therapy, whereas HER2 positive IBC benefits from anti-HER2 therapy. IBC that (over-)expresses neither HR nor HER2 [triple negative breast cancer (TNBC)] typically responds well to chemotherapy, which is mostly administered in the neoadjuvant setting [8]. 

In 2000 and 2001, gene expression analysis led to the identification of IBC molecular subtypes, namely Luminal subtypes, HER2-enriched, Basal-like, and normal breast-like [9,10,11]. The identified molecular subtypes were then revised into Luminal A, Luminal B, HER2-enriched and Basal-like, given their prognostic and predictive value used for therapy recommendation [12]. Patients with Luminal A IBC have a very good prognosis and benefit from endocrine therapy alone but not from chemotherapy in a clinically relevant dimension. However, due to endocrine resistance, patients with Luminal B tumors might have a poorer prognosis if treated with endocrine therapy alone [13] but might profit from chemotherapy [9,10,12]. HER2-enriched IBC are highly sensitive to anti-HER2 agents [14]. Finally, patients with Basal-like IBC benefit from chemotherapy [15]. 

In daily routine diagnostics, surrogate (molecular-like) IBC subtyping is assessed using the immunohistochemical (IHC) expression of ER and PR, HER2 status [IHC and/or in situ hybridization (ISH)], the IHC expression of the proliferation marker Ki-67, and, depending on the classification definition used, tumor grade. Luminal A-like IBCs show expression of HRs, no overexpression of HER2, low Ki-67 expression (and low/intermediate tumor grade). Luminal B-like HER2 negative tumors display HR positivity, HER2 negativity but high Ki-67 (and/or high-grade morphology), whereas Luminal B-like HER2 positive IBCs express both HRs and HER2 independently from the proliferation rate (tumor grade low, intermediate or high). HER2 positive (non-luminal) IBCs show HER2 positivity but HR negativity. As mentioned above, TNBCs display both negative HR and HER2 status [16,17].

In patients with HR+/HER2- IBC of intermediate risk of recurrence, estimated using conventional clinical and pathological risk factors, the decision on adjuvant chemotherapy is very challenging for clinicians. In order to address this issue, several multigene expression tests have been developed to assess the risk of distant recurrence and, in part, to evaluate the molecular subtypes which, however, differ from test to test due to the different technologies and the gene expression profiles used [11]. These tests include Oncotype DX® (Genomic Health), MammaPrint® (Agendia), EndoPredict® (Sividon Diagnostics/Myriad Genetics), Prosigna®, (Veracyte, formerly: NanoString Technologies)] [8]. Formalin-fixed and paraffin-embedded (FFPE) tumor tissue can be used for all assays. All these multigene expression tests provide both risk scores and discrimination into risk groups. They also provide information about late recurrence (EndoPredict® and Prosigna®) and, partly, even information about the molecular subtype (Prosigna®, Blueprint® if added to MammaPrint®). So far, prospective studies have shown that Oncotype DX® and MammaPrint® reveal patients at low risk of recurrence that would be overtreated with chemotherapy [18,19,20]. Currently, there are no available results from prospective trials regarding the predictive value of EndoPredict® and Prosigna® [21]. However, retrospective studies provide strong evidence that the risk scores of these two gene expression tests predict well both late distant recurrence and patients at low risk [22]. Disadvantages of all multigene expression assays are the high costs (~2–3 k Euro in Germany) and the fact that these tests are not comprehensively available in pathological laboratories. 

Given the ability of Prosigna® test to provide both a risk of recurrence and a PAM50-based IBC molecular subtype, the present multicenter study aims to investigate whether Prosigna® assay results [testing centers comprised the Institutes of Pathology of Erlangen [coordinating center, C1), München (C2), Viersen (C3), Halle/Saale (C4), and Essen (C5)] correlate with the molecular-like surrogate subtypes, routinely assessed [locally vs. by C1 vs. by the study site Salzburg (C6)] using immunohistochemistry (+/− tumor grade). Furthermore, the impact of Prosigna® test results on treatment decision is investigated.

## 2. Results

Prosigna® molecular subtyping was compared with local and two central (C1 and C6) molecular-like subtypes relying on both IHC (HRs, HER2, Ki-67) subtyping and IHC + tumor grade (IHC+G) subtyping. Furthermore, the influence of Prosigna® assay results on chemotherapy treatment decision was investigated. 

### 2.1. Characteristics of Cohort

Prognostic clinical and pathological variables including age at diagnosis, tumor size, Ki-67 expression, and tumor grade are summarized in Appendix A. Briefly, median patients’ age at diagnosis was 55 years (range: 39–78 years); tumor size ranged between 0.7 and 15.4 cm (median 1.8 cm). There was a significant association between assessment center (local, C1, or C6) and tumor grade (*p*-value = 1.6 × 10^−9^) (Appendix A). Notably, center C6 was positively associated with G1 tumors (21/109) and negatively associated with G3 tumors (7/109) (Appendix A). Furthermore, there was a significant difference between the three centers also in Ki-67 expression (*p*-value = 5.8 × 10^−6^). Indeed, while local and C1 Ki-67 assessments did not show any significant difference in terms of median expression value, the one reported by center C6 was significantly lower compared to both center C1 (post-hoc pairwise adjusted *p*-value = 1.2 × 10^−4^) and local institutions (post-hoc pairwise adjusted *p*-value = 1.4 × 10^−5^) (Appendix A). Taken together, these results show that in the evaluation of some prognostic clinical and pathological variables, center C6 performed differently from both local institutions and center C1, which instead provided comparable results. 

### 2.2. Comparison of Local versus Central Molecular-Like Subtyping

According to local IHC subtyping, 35.4% of cases were Luminal A-like and 64.6% Luminal B-like subtype, whereas according to IHC+G subtyping 31.9% and 68.1% were Luminal A-like and Luminal B-like, respectively. When analyzing both C1 IHC and IHC+G subtyping, the proportion of the different subtypes was comparable to the one reported from local assessments, with Luminal B being the dominant surrogate subtype. In center C6, however, the opposite scenario was observed, with Luminal A-like subtype accounting for more than two thirds of all the cases (Table 1). 

The difference between the two centers C1 and C6 in the proportion of surrogate subtypes was further explored. Almost all IHC+G cases declared as Luminal A-like by C6 but not by C1 reported in C1 Ki-67 values ≥ 20% and were thus classified as Luminal B-like (HER2 negative) tumors (Appendix A). Instead, the IHC+G cases declared as Luminal B-like HER2- by C1, but classified as Luminal A-like by C6, were characterized by Ki-67 values < 20% according to C6 (Appendix A). Among the IHC+G Luminal A-like cases assessed in center C6, 26.9% (18/67) were graded as G1, whereas the remaining 73.1% (49/67) were of intermediate grade (G2). Only 5.6% (1/18) of the cases assessed as G1 Luminal A-like by C6 were graded as G1 also by C1, while almost all (94.4%, 17/18) were graded as G2. Of these, 76.5% (13/17) were still Luminal A-like cases but 23.5% (4/17) were upgraded to Luminal B-like HER2 negative (Appendix A) due to Ki-67 expression ≥ 20%. A total of 73.5% (36/49) of G2 Luminal A-like cases detected by C6 were graded as G2 also by C1. Of these, 63.9% (23/36) were still Luminal A-like, 30.5% (11/36) were upgraded to Luminal B-like HER2 negative, and 5.6% (2/36) were Luminal B-like HER2 positive. The remaining 26.5% (13/49) cases were upgraded to G3 by C1 and all classified as Luminal B-like HER2 negative (Appendix A).

### 2.3. Distribution of Prosigna® Molecular Subtypes

A total of 53.8% (n = 64/119) of cases were allocated as Prosigna® Luminal A and 44.5% (n = 53/119) as Prosigna® Luminal B. Two cases, however, did not match the local HR positivity (both were locally assessed as Luminal B-like HER2 negative) and were assigned to Basal-like (n = 1/119) and HER2-enriched (n = 1/119) Prosigna® subtypes.

### 2.4. Comparison between Surrogate Subtyping and Molecular Prosigna® Subtype

In all, 82.5% (33/40) of local IHC Luminal A-like tumors were classified as Prosigna® Luminal A. However, 17.5% (7/40) were upgraded to Prosigna® Luminal B. Within local IHC Luminal B-like IBCs, 58.9% (43/73) matched with Prosigna® Luminal B subtype, whereas 38.4% (28/73) were downgraded to Prosigna® Luminal A (Figure 1A). According to local IHC+G subtyping, 83.3% (30/36) of Luminal A-like cases were classified by Prosigna® as Luminal A and 16.7% (6/36) were upgraded to Prosigna® Luminal B. In regard to local IHC+G Luminal B-like IBCs, 57.1% (44/77) were classified as Prosigna® Luminal B, whereas 40.3% (31/77) were downgraded to Prosigna® Luminal A (Figure 1B). 

Compared with C6 assessments, local and C1 subtyping showed less cases upgrading from Luminal A-like to Prosigna® Luminal B (Table 2). With respect to center C6, though, they reported more IBC cases downgrading from Luminal B-like to Prosigna® Luminal A (Table 2, Figure 1). 

The best strength of agreement occurred between Prosigna® and center C1 for both IHC and IHC+G subtyping, with a Cohen’s Kappa (κ) of 0.449 and 0.455, respectively. When instead considering the degree of agreement between Prosigna® and center C6, the k value reached 0.379 for IHC subtypes and 0.36 for IHC+G subtypes (Table 3).

### 2.5. Prosigna® Risk Groups and Correlation with Local Surrogate Subtypes

A total of 87.5% (56/64) of all Prosigna® Luminal A tumors showed a risk of recurrence (ROR) < 50%, whereas 92.5% (49/53) of all Prosigna® Luminal B tumors presented with a ROR ≥ 50% (Figure 2A). Furthermore, there was a significant association (*p*-value = 3.8 × 10^−14^) between Prosigna® subtypes and Prosigna® risk groups (low, intermediate, and high). Namely, 82% (41/50) of patients at high risk had Prosigna® Luminal B IBC subtype, whereas 71.1% (27/38) of patients at intermediate risk and 93.5% (29/31) at low risk suffered from Prosigna® Luminal A tumors (Figure 2B).

Prosigna® risk group was found to be significantly correlated also with both IHC and IHC+G subtypes (Figure 3), with Prosigna® high risk group being positively associated with Luminal B subtype and negatively associated with Luminal A subtype (*p*-value = 0.007 and *p*-value = 0.015, respectively, for IHC and IHC+G subtypes).

### 2.6. Influence of Prosigna® Assay Result on Treatment Recommendation

Looking at the distribution of tumor board recommendation within the different Prosigna® subtypes, chemotherapy + endocrine therapy was recommended to 44.2% (23/52) of Prosigna® Luminal A cases and to 88.6% (39/44) of Prosigna® Luminal B cases. The two cases with Prosigna® Basal-like (n = 1) and Prosigna® HER2-enriched (n = 1) IBC were both recommended towards adjuvant chemotherapy + endocrine therapy (Figure 4).

Within local IHC surrogate subtyping, 51.6% (16/31) of Luminal A-like IBCs were recommended towards chemotherapy + endocrine therapy. All these cases were indeed classified by Prosigna® as either at high or intermediate risk (Figure 5A). Notably, looking at the distribution of Prosigna® subtypes within these Luminal A-like IHC cases, 68.8% (11/16) were Prosigna® Luminal A subtypes at high or intermediate risk and the remaining 31.3% (5/16) were upgraded to Prosigna® Luminal B at high or intermediate risk (Figure 5B). 

Within local IHC+G subtyping, 44.4% (12/27) Luminal A-like tumors were recommended towards chemotherapy + endocrine therapy (Figure 5C). In all, 75% (9/12) of these were classified as Luminal A subtypes at high or intermediate risk group according to Prosigna® assay, whereas the remaining 25% (3/12) were upgraded to Prosigna® Luminal B subtype at high risk (Figure 5D).

## 3. Discussion

In the present study, we correlated the results of a PAM50-based multigene expression test (Prosigna®) with surrogate subtyping of 119 IBC patients and investigated the influence of Prosigna® results on therapy decision. We showed that the agreement between Prosigna® molecular subtypes and molecular-like subtyping using IHC +/− tumor grade was fair to moderate depending on surrogate subtyping method and center. For Luminal A-like cases locally assessed either with IHC or IHC+G subtyping, results showed that adjuvant chemotherapy + endocrine therapy was recommended by the interdisciplinary tumor board to those cases that Prosigna® testing classified as Luminal A at high/intermediate risk or upgraded to Luminal B.

Commercially available breast cancer multigene expression tests (e.g., Oncotype DX®, MammaPrint®, EndoPredict®, Prosigna®) have been developed to help oncologists in the decision for or against adjuvant chemotherapy in patients with early HR+ IBC at intermediate risk. In prospective–retrospective (Oncotype DX®, EndoPredict®, Prosigna®) and prospective (Oncotype DX®, MammaPrint®) trials, respectively, it has been shown that multigene expression tests can identify IBC patients with low risk of recurrence that can be treated with adjuvant endocrine therapy but do not need additional chemotherapy [13,18,19,20,23,24,25,26,27,28,29,30,31,32,33,34]. However, application of these tests should be restricted to a narrow IBC patient cohort in which the use of multigene expression tests is reasonable [35,36]. 

In detail, the Prosigna® assay was validated to predict the outcome in (a) post-menopausal women with HR+, lymph node-negative, Stage I or II IBC and (b) post-menopausal women with hormone receptor-positive (HR+), lymph node-positive (1–3 positive nodes), Stage II IBC after standard of care loco-regional treatment and adjuvant endocrine therapy alone. If used in conjunction with other clinical and pathological risk factors, the Prosigna® assay may provide additional information about the probable distant recurrence-free survival at 10 years [37]. Indeed, besides the PAM50-based molecular subtype, Prosigna® assay predicts the risk of distant recurrence within 10 years and the risk group [38]. In our study of HR+ (locally assessed) IBC cases, 53.8% were classified as Prosigna® Luminal A subtype and 44.5% as Prosigna® Luminal B subtype. Two outlier cases were found, one Basal-like and one HER2-enriched. Both were locally classified as Luminal B-like HER2 negative. Within the Prosigna® high risk group, most cases were of the Prosigna® Luminal B subtype; vice versa, almost the entire low risk group was characterized by Prosigna® Luminal A IBCs. This is in line with both the biological understanding of molecular subtyping of IBC and its prognostic and predictive value. Indeed, Luminal A tumors are associated with a better prognosis compared to Luminal B IBCs; hence, therapy stratification can be modified accordingly [8]. As mentioned above, surrogate IBC subtyping using IHC +/- tumor grade has been used in daily routine diagnostics instead of mRNA-based molecular subtyping for the last decades. Indeed, compared to multigene expression assays, surrogate subtyping benefits from several advantages: it is cheaper, of low turn around time, and available even in small pathological laboratories. However, as highlighted by our study, the expression of single biomarkers (e.g., Ki-67) can differ from site to site due to pre- and post-analytical reasons [39,40]. Hence, oncologists have to consider the (post-) analytical standards of their local pathologists for therapy stratification. Notably, when comparing local and two central (C1, C6) assessments of surrogate subtypes, differences emerged between center C6 and both the local and C1 institutions. Indeed, while both local and C1 laboratories detected a higher proportion of Luminal B-like cases compared to Luminal A-like cases, in center C6 the majority of samples were classified as Luminal A-like. This is of great importance, since therapy stratification may vary. Discrepancies in surrogate subtyping were due to significant differences in both Ki-67 expression values and tumor grade assessments. These findings confirm that, to date, distinction between Luminal A-like and Luminal B-like tumors by IHC is still problematic and controversial. Some people suggest emphasizing tumor grade and PR expression in regard to luminal subtype distinction [17,41,42], while others advise the use of Ki-67 IHC [16]. Although big standardization efforts have been made [39,43,44], there is still a lack of common internationally accepted guidelines for the use, standardized scoring method, and optimal cut off for Ki-67 expression [16,39,45,46]. One should note that the use of Ki-67 IHC for therapy decision depends on (inter-)national guidelines: It is recommended by the St. Gallen International Expert Consensus but not by the American Society of Clinical Oncology [16,46,47,48]. 

When comparing molecular-like subtypes with Prosigna® subtyping, we found a match up to 86.8% for Luminal A(-like) cases and up to 75.9% for Luminal B(-like) cases depending on center and surrogate subtyping method. Accordingly, a fair-to-moderate agreement was found between Prosigna® molecular subtypes and molecular-like subtyping. This is in line with the findings of Bastien et al., who reported some inconsistency between IHC surrogate subtyping and PAM50 gene expression subtypes. In their study, *ESR1* and *ERBB2* gene expression showed more prognostic impact than the corresponding IHC markers [49]. In another study that correlated IHC-based surrogate subtyping with PAM50 molecular subtypes, 38.4% of IBCs were discrepantly subtyped [50]. Furthermore, agreement between PAM50 HER2 enriched tumors and HER2 positive subtype defined by standard IHC/ISH is not always given [51,52], which might lead to confusion and fundamental changes in regard to therapy recommendation. In a TNBC cohort, 84.3% of cases matched with PAM50 Basal-like subtype, 16.7% were HER2-enriched, and 5.2% showed a luminal gene signature (4.2% Luminal A, 1.0% Luminal B) [53].

In our study, the distribution of Prosigna® risk groups within local surrogate subtypes (both IHC and IHC+G) highlighted a significant association between Prosigna® high risk tumors and local Luminal B-like subtyping, whereas local Luminal A-like IBCs positively associated with Prosigna® low risk cases, which is in line with the expected biological behaviour of surrogate subtypes.

When comparing IHC / IHC+G molecular-like subtypes with Prosigna® molecular subtypes, 13.2% to 31.3% of Luminal A-like tumors were upgraded to Prosigna® Luminal B, whereas 24.1% to 40.3% of Luminal B-like IBCs were downgraded to molecular Luminal A. Especially for Luminal A-like cases that are classified as Prosigna® Luminal B, this upgrade may influence oncologists’ decision on additional chemotherapy to adjuvant endocrine therapy. In our study, Luminal A-like tumors that were recommended to be treated with chemotherapy were either of higher risk and/or upgraded to Prosigna® Luminal B. Hence, the combination of Prosigna® molecular subtyping and risk estimation may have an impact on therapy stratification, especially for HR+ HER2- IBC patients with intermediate Ki-67 level / tumor grade.

A notable limitation of this study is the relatively small number of cases. A total of 18 samples had indeed to be excluded due to missing surrogate subtyping. The lack of clinical and pathological parameters was mainly attributable to those cases for which routine diagnostic was performed in laboratories not offering Prosigna® testing assay and that turned to central institutes (i.e., C1-C5) for gene expression testing.

Regarding the decision for or against chemotherapy, only the interdisciplinary tumor board decision after multigene expression assay performance was available. Hence, we could describe the (surrogate) subtypes and risk groups that were recommended to be treated with chemotherapy + endocrine therapy. However, we were not able to track any changes, either in gynecological oncologists-recommended therapy or patient’s attitude, in favor of adding adjuvant chemotherapy. To address this issue, an ongoing single-center study (C1) is currently evaluating prospective questionnaires to assess patients’ decision about adding adjuvant chemotherapy before and after molecular testing. Furthermore, due to missing survival information and the small number of cases, we could not correlate Prosigna® risk estimation with patients’ outcome and menopausal status. Therefore, we could analyze which surrogate subtyping matches best with Prosigna® subtyping but not which subtyping method (IHC vs. IHC+G, local vs. central, surrogate vs. molecular subtyping) has the greatest impact on prognosis.

Although the multigene expression assays mentioned above provide independent, and partly similar predictions on prognosis, the various tests accurately described by Sinn et al. cannot be compared directly with each other. Indeed, there is a considerable “inter-assay heterogeneity” which includes variability in test development, different test measures, only partly overlapping gene signatures, and variable risk scores [11,54,55]. Furthermore, classifying special histological types of IBC (e.g., adenoid-cystic breast carcinoma, secretory breast carcinoma) using gene expression tests without closer examination of underlying mechanisms may not reflect the distinct biology and outcome, and may thus require additional investigation [56].

In intermediate risk hormone receptor (HR) positive, HER2 negative breast cancer (BC), the decision regarding adjuvant chemotherapy might be also supported by using the PAM50-based genomic signature, the chemoendocrine score (CES), which is predictive of poorer relapse-free survival in patients with ROR-intermediate IBC treated with either adjuvant endocrine therapy only or no adjuvant systemic therapy, but not in patients treated with adjuvant chemotherapy [57]. Unfortunately, since PAM50-based molecular subtype in our cohort was performed by using the commercial Prosigna® assay, we were not able to calculate the CES values which might have added further valuable information on prognosis in our cases with the Prosigna® intermediate risk group.

McVeigh reported that the use of Oncotype DX influenced the choice of therapy, leading to 57% of the patients being spared from chemotherapy [58]. Similarly, we demonstrated that 48.4% of patients with surrogate Luminal A-like tumors at intermediate risk were recommended to receive endocrine therapy alone. Several decision impact and cost-effectiveness analyses have been conducted, showing multigene expression testing being cost-effective or one being more effective than another test [59,60,61,62]. One of the main disadvantages of these impact studies, however, is mostly the lack of integration with patients’ outcome [54].

## 4. Materials and Methods

### 4.1. Study Design

A total number of n = 142 IBC cases were prospectively included into the study between 2015 and 2016 and analyzed for further risk stratification within diagnostic setting using multigene expression testing (Prosigna® assay, to date of testing: NanoString Technologies, Seattle, WA, U.S.A.) at the Institutes of Pathology of Erlangen (coordinating center, C1), München (C2), Viersen (C3), Halle/Saale (C4), and Essen (C5). Histopathological diagnoses of invasive breast cancer (IBC) were made either in one of the aforementioned centers or decentralised in a peripheral pathological laboratory according to the German guidelines and recommendations relevant during those years [63]. Clinical and pathological data (e.g., age, histological subtype, tumor size, nodal status, tumor grade, IHC expression of ER, PR, and Ki-67, and HER2 status) were collected and, whenever available, also local tumor board’s therapy recommendations from original patients’ records and pathological reports. For all decentral pathological laboratories not offering Prosigna® testing, gene expression analysis was performed by one of the five study centers. All cases with sufficient formalin-fixed and paraffin-embedded (FFPE) tissue availability underwent further testing in center C1 and study site Salzburg (C6). These two centers retrospectively performed a second “central” assessment of tumor grade and IHC / ISH (ER, PR, Ki-67, HER2). Within each of the two centers C1 and C6, two pathologists, blinded to local data and experienced in breast cancer pathology, evaluated the surrogate subtypes. In case of disagreement between the two pathologists of the same center, consensus was reached after having reviewed together the respective assessments. Details in tissue processing for routine diagnostics, immunohistochemistry, and HER2 chromogenic in situ hybridization (CISH) can be found in the Supplement. Approval of the local academic ethics committee of the Friedrich-Alexander-Universität Erlangen-Nürnberg was obtained. 

### 4.2. Molecular-Like Surrogate Subtyping

Based on local and central (C1 and C6) pathology, respectively, which involves the assessment of IHC expression of ER, PR, Ki-67 as well as HER2 status, and tumor grade, IBC samples were classified in surrogate subtypes according to: 

(1) IHC (+/− ISH) alone (“IHC subtyping”; local vs. C1 vs. C6) (Table 4);

(2) IHC (+/− ISH) and tumor grade (”IHC+G subtyping”; local vs. C1 vs. C6) (Table 5). 

### 4.3. Prosigna® Assay

The Prosigna® test (Veracyte, South San Francisco, CA, USA; formerly: NanoString Technologies, Seattle, WA, USA) was established in our laboratories between 2014 and 2016. For establishment, inter-laboratory agreement on n = 15 IBC cases (Institutes of Pathology Erlangen, München, and Heidelberg) was evaluated. Both tumor molecular subtype estimation and risk group assessment correlated in 100% of samples. Before RNA isolation, experienced pathologists reviewed the IBC cases regarding adequate tumor tissue on H&E slides, marked tumor region, and assessed tumor content on each tumor slide. RNA extraction and the multigene expression assay were performed by trained technicians according to the manufacturer’s manual. Assay measurements were done fully automatically by nCounter® Prep Station and nCounter Dx Digital Analyzer. The report file in output provided information about the molecular subtype, risk of recurrence (ROR), risk group (low, intermediate, or high), and probability of distant recurrence for each patient [65].

### 4.4. Statistical Analysis

All statistical analyses were performed within the R environment v.4.0.3 [66]. *p*-values (or, where applicable, adjusted *p*-values) < 0.05 were considered statistically significant. Details can be found in the Appendix A.

## 5. Conclusions

In conclusion, we could demonstrate that IHC/IHC+G surrogate subtyping and molecular Prosigna® subtyping correlate moderately. The best pairwise agreement occurred, for both IHC and IHC+G, between Prosigna® and central C1 surrogate subtyping. Local Luminal A-like cases (IHC/IHC+G) were recommended towards chemotherapy + endocrine therapy if they were classified as Prosigna® Luminal A at high/intermediate risk or upgraded to Prosigna® Luminal B. Hence, the additional use of molecular subtyping and risk profiling might influence therapy recommendation or confirm oncologists’ choice of therapy in HR+ HER2- IBCs with intermediate prognostic risk assessed using conventional clinical and pathological risk factors.

## Figures and Tables

**Figure 1 ijms-23-08716-f001:**
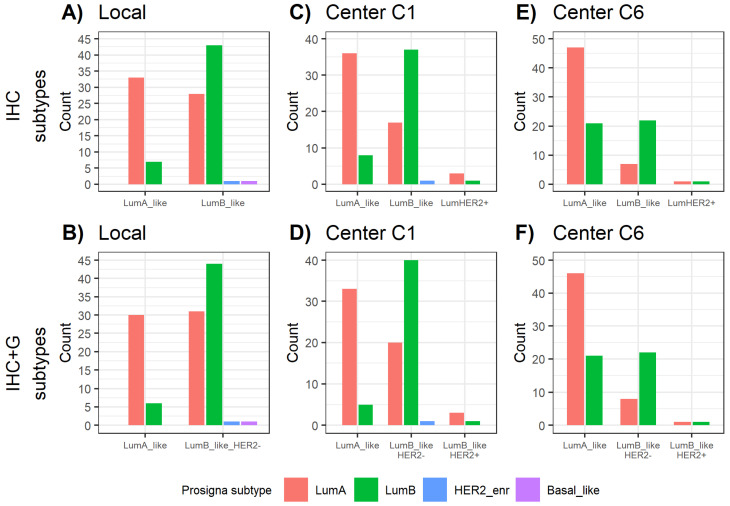
Distribution of Prosigna® subtypes across (**A**) local IHC subtypes, (**B**) local IHC+G subtypes, (**C**) C1 IHC subtypes, (**D**) C1 IHC+G subtypes, (**E**) C6 IHC subtypes, (**F**) C6 IHC+G subtypes. HER2_enr = HER2-enriched; IHC = immunohistochemistry; IHC+G = immunohistochemistry + tumor grade; LumA = Luminal A; LumB = Luminal B.

**Figure 2 ijms-23-08716-f002:**
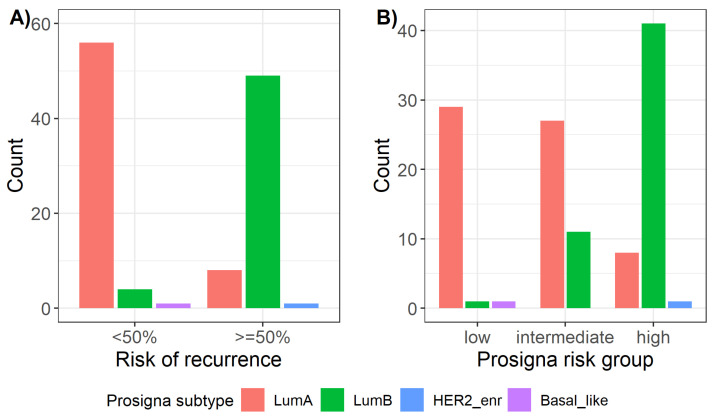
Distribution of Prosigna® subtypes across (**A**) Prosigna® risk of recurrence and (**B**) Prosigna® risk groups. HER2_enr = HER2-enriched; LumA = Luminal A; LumB = Luminal B.

**Figure 3 ijms-23-08716-f003:**
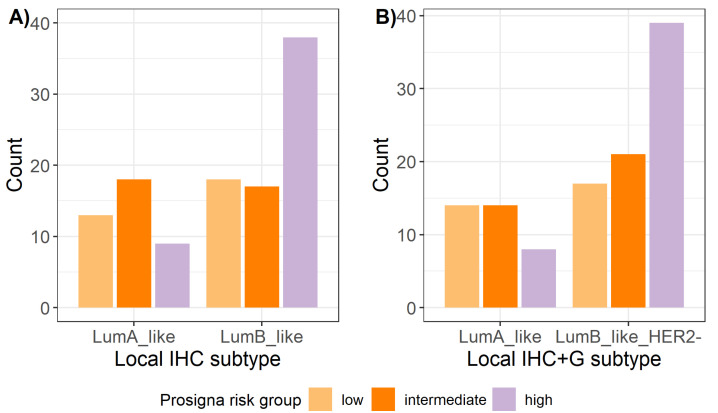
Distribution of Prosigna® risk groups across local surrogate (**A**) IHC subtypes and (**B**) IHC+G subtypes. IHC = immunohistochemistry; IHC+G = immunohistochemistry + tumor grade; LumA = Luminal A; LumB = Luminal B.

**Figure 4 ijms-23-08716-f004:**
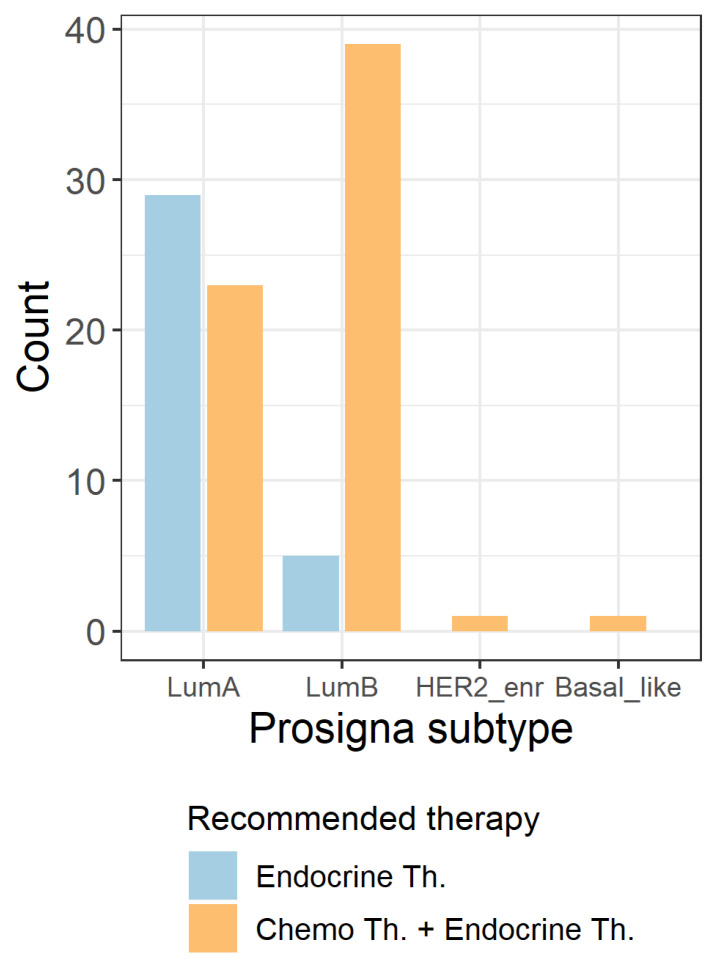
Distribution of tumor board recommended across the different Prosigna® subtypes. HER2_enr = HER2-enriched; LumA = Luminal A; LumB = Luminal B; Th. = therapy.

**Figure 5 ijms-23-08716-f005:**
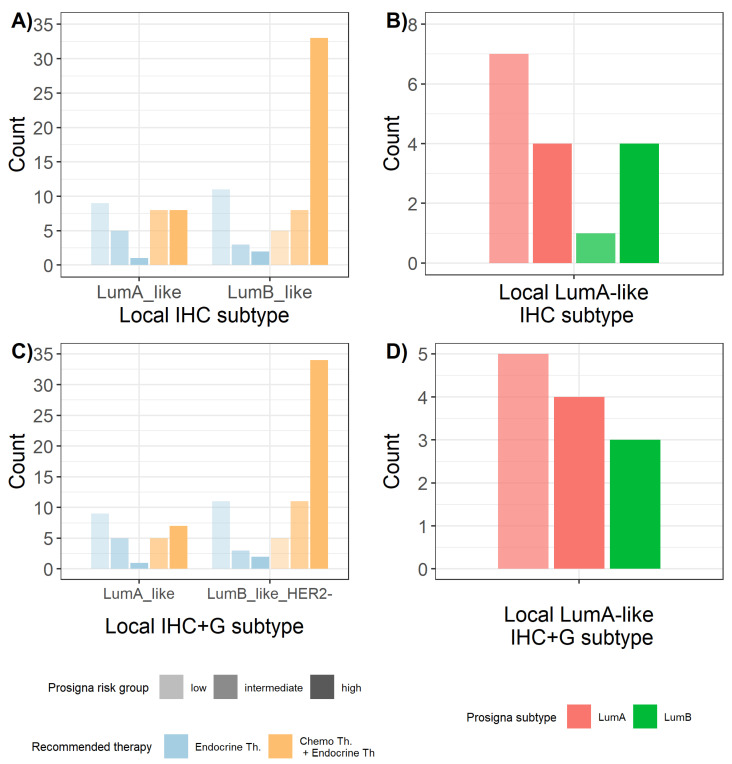
Influence of Prosigna® results on treatment decision for locally assessed IHC and IHC+G surrogate subtypes. (**A**) Recommended therapy within locally assessed IHC surrogate subtypes and (**B**) distribution of Prosigna® subtype within the subset of Luminal A-like IHC subtypes that were recommended towards adjuvant chemotherapy and endocrine therapy. (**C**) Recommended therapy within locally assessed IHC+G surrogate subtypes and (**D**) distribution of Prosigna® subtype within the subset of Luminal A-like IHC+G subtypes that were recommended towards adjuvant chemotherapy and endocrine therapy. Colour intensity reflects Prosigna® risk group. IHC = immunohistochemistry; IHC+G = immunohistochemistry + tumor grade; LumA = Luminal A; LumB = Luminal B; Th. = therapy.

**Table 1 ijms-23-08716-t001:** Frequency of the different surrogate subtypes locally and centrally (C1, C6) assessed using both immunohistochemistry (IHC) and IHC+G subtyping.

	Surrogate Subtypes		
	Luminal A-like	Luminal B-like HER2 negative	Luminal B-like HER2 positive
Local IHC subtyping	35.4%	64.6%	//
Local IHC+G subtyping	31.9%	68.1%	//
C1 IHC subtyping	42.7%	53.4%	3.9%
C1 IHC+G subtyping	36.9%	59.2%	3.9%
C6 IHC subtyping	68.7%	29.3%	2.0%
C6 IHC+G subtyping	67.7%	30.3%	2.0%

HER2: human epidermal growth factor receptor 2.

**Table 2 ijms-23-08716-t002:** Comparison between local vs. C1 vs. C6 surrogate subtyping and molecular Prosigna® subtype.

	Luminal A-Like		Luminal B-Like	
	Match with Prosigna® Luminal A	Upgrade to Prosigna® Luminal B	Match with Prosigna® Luminal B	Downgrade to Prosigna® Luminal A
Local IHC subtyping	82.5% (33/40)	17.5% (7/40)	58.9% (43/73)	38.4% (28/73)
Local IHC+G subtyping	83.3% (30/36)	16.7% (6/36)	57.1% (44/77)	40.3% (31/77)
C1 IHC subtyping	81.8% (36/44)	18.2% (8/44)	67.3% (37/55)	30.9% (17/55)
C1 IHC+G subtyping	86.8% (33/38)	13.2% (5/38)	65.6% (40/61)	32.8% (20/61)
C6 IHC subtyping	69.1% (47/68)	30.9% (21/68)	75.9% (22/29)	24.1% (7/29)
C6 IHC+G subtyping	68.7% (46/67)	31.3% (21/67)	73.3% (22/30)	26.7% (8/30)

IHC: immunohistochemistry.

**Table 3 ijms-23-08716-t003:** Estimates of Cohen’s Kappa (κ) as an index of pairwise agreement between Prosigna® subtypes and local assessments (IHC / IHC+G subtypes), Prosigna® subtypes and C1 assessments (IHC / IHC+G subtypes), Prosigna® subtypes and C6 assessments (IHC / IHC+G subtypes).

	κ (IHC Subtype)	κ (IHC+G Subtype)
Prosigna vs. local institutes	0.374	0.344
Prosigna vs. center C1	0.449	0.455
Prosigna vs. center C6	0.379	0.36

IHC: immunohistochemistry.

**Table 4 ijms-23-08716-t004:** Criteria for surrogate IHC subtyping according to [16].

Surrogate Subtype	Subgroup	ER	PR	HER2		Ki-67 (%)
Luminal A-like		+	+/−	−	and	Low (<20%)
Luminal B-like	HER2 negative	+	+/−	−	and	High (≥20%)
	HER2 positive	+	+/−	+		Any value
HER2 positive (non-luminal)		−	−	+		Any value
Triple negative		−	−	−		Any value

ER: estrogen receptor; HER2: human epidermal growth factor receptor 2; IHC: immunohistochemistry; Ki-67: proliferation marker; PR: progesterone receptor.

**Table 5 ijms-23-08716-t005:** Criteria for surrogate IHC+G subtyping according to [16,17,64].

Surrogate Subtype	Subgroup	ER	PR	HER2		Tumor Grade		Ki-67 (%)
Luminal A-like		+	+/−	−	and	G1, G2	or	Low (<20%)
Luminal B-like	HER2 negative	+	+/−	−	and	G3	or	High (≥20%)
	HER2 positive	+	+/−	+		G1, G2, G3		Any value
HER2 positive (non-luminal)		−	−	+		G1, G2, G3		Any value
Triple negative		−	−	−		G1, G2, G3		Any value

ER: estrogen receptor; HER2: human epidermal growth factor receptor 2; IHC: immunohistochemistry; Ki-67: proliferation marker; PR: progesterone receptor.

## Data Availability

Detailed data not shown in the manuscript are available on request.

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
