# Peer review of "Molecular Subtyping of Invasive Breast Cancer Using a PAM50-Based Multigene Expression Test-Comparison with Molecular-Like Subtyping by Tumor Grade/Immunohistochemistry and Influence on Oncologist’s Decision on Systemic Therapy in a Real-World Setting"

_ijms, 2022, doi:10.3390/ijms23158716_

Round 1

Reviewer 1 Report

The authors provide an interesting retrospective study regarding the correlation between breast cancer immunohistochemistry classification and molecular subtype classification with Prosigna, analysing implications in therapeutic-decision making.  I have some minor concerns that I would like the authors to address:

1) In the Introduction, at page 2, lines 86-89, the authors mentioned all intrinsic subtypes and their respective treatment sensitivities, except for HER2-Enriched tumors. A resumen of this subtype’s characteristics can be found at PMID 34392185. For the purpose of this sentence I would suggest to simply state that HER2 enriched tumors are highly sensitive to antiHER2 agents. 

2) Subtypes identified through Blueprint/Mammaprint are different biomarkers than intrinsic subtypes identified via PAM50/Prosigna. These are technically and biologically differing clinical entities and should be thus avoided any confusion. For a clear explanation please refer to PMID 35853907. In this perspective I would suggest to avoid the term of intrinsic subtype and rather use the terminology “molecular subtype”, which in this case seem to be more appropriate and comprehensive. 

3) I had difficulties in following the C1-C6, local/centralized assessment thing. I understand that maybe due to journal formatting issues, the Methods are reported at the end of the paper, but could it be possible to make some reference to the testing centers in the Intro or in the Results starting section? At present results are hard to follow. 

4) Can the authors clarify what they mean with the sentence at page 12 line 369-370? Are they making reference to different molecular subtyping (Blueprint vs Prosigna vs Research-base PAM50) or differences in molecular risk stratification?

5) The CES score (PMID: 27903675) might be a useful tool for intermediate risk score cases. Unfortunately it has not been introduced in clinical practice so far, despite it could be in theory obtained by the Prosigna test itself. The authors might mention this tool in the Discussion and put it in the context of their results. 

6) Considering that the casuistry is referred to 2015-2016, why the authors did not provide also some prognostic/predictive information? It might be a useful addition, at least in the supplementary materials (since I understand that this was not the focus of the paper).

Author Response

Dear Reviewer,

we highly appreciate your time and thank you a lot for your valuable comments. According to your recommendations, we have modified the manuscript.

1) We totally agree and have added both an explanation in regards to therapy options of HER2-enriched invasive breast cancer (page 2, line 89) and the reference suggested by the reviewer.

2) We thank the reviewer for this helpful comment. A statement that different multigene expression tests deliver different molecular subtypes (page 3, lines 106-107) and the corresponding reference suggested have been added (page 2, line 83 and page 3, lines 106-107). Moreover, we have replaced “intrinsic subtype” by “molecular subtype” in both the manuscript and the supplement.

3) We fully understand the point made by the reviewer and have added an explanation of the study centers involved at the end of the introduction (page 3, lines 124-127).

4) We apologize for this misleading sentence. Since it is both misleading and redundant to the statement before, we have deleted this sentence. However, we have added PMID 35853907 to the previous sentence (pages 11-12, lines 371-373).

5) We highly appreciate this valuable comment on the CES which we have included into the discussion and put it into the context of our analyses (page 12, lines 378-386).

6) Being very aware of the clinical relevance and importance of correlation of our results with clinical data, we have to admit that, unfortunately, we do not have access to most of the survival data of our cohort. The lack of clinical parameters is mainly attributable to those cases for which routine diagnostic was performed in laboratories not offering Prosigna® testing assay and that turned to central institutes (i.e. C1-C5) for gene expression testing only. I.e., most patients were diagnosed and treated at smaller decentral sites with the pathological laboratories not being able to perform the Prosigna® test by themselves. If the clinicians of this hospitals wished for Prosigna® testing, a suitable FFPE block plus information on tumor size and nodal status was delivered to the study sites C1-C5 but not any further clinical data. In addition, although Prosigna® examination was sometimes ordered as an inpatient procedure, further treatment/follow-up was provided by outpatient oncologists who were not known to the decentralized pathologists. Therefore, it was almost impossible to obtain more clinical information about the patients in our cohort (briefly mentioned on page 11, lines 350-353) and reliable statistical analyses on the few cases with clinical data not meaningful, respectively.

Best regards.

Reviewer 2 Report

In this manuscript, Ramona Erber and colleagues performed a prospective multicentric study for 142 intermediate risk breast cancer patients using a proposed Prosigna PAM50-based multigene expression test. The authors compared Prosigna molecular subtyping with local and two central (C1 and C6) molecular-like subtypes based on both IHC and IHC + tumor grade surrogate subtyping. They observed the best agreement between Prosigna molecular subtyping with C1 surrogate subtyping. In addition, the authors investigated the influence of Prosigna assay results on chemotherapy treatment decision, and they further provide some recommendations. Overall, this manuscript that is well-written, and will definitely help the oncologists to make clinical decision that the patients will benefit. The authors, however, should consider the following specific comments to further strengthen the manuscript.

1\ The author should also compare Prosigna molecular subtyping with other subtypes based on IHC in several tables and Figures, especially triple negative breast cancer (TNBC), which is the most malignant breast tumor.

2\ Very curious about if the oncologists have applied the Prosigna molecular subtyping and related treatment decision on the clinical, if yes, the author should add some results about if the chemotherapy treatments based on the new subtyping have better outcome?

Author Response

Dear Reviewer,

We thank you for your positive feedback and your comments and constructive criticisms.

1) We share the opinion of the reviewer that a comparison between Prosigna® molecular subtyping and IHC-based triple-negative breast cancer in a real-world cohort, i.e. patients with invasive breast cancer being diagnosed and treated not within a clinical study but in daily routine operations, is very interesting and clinically relevant. However, our cohort was focused on patients with hormone receptor positive/HER2 negative invasive (HR+/HER2-) breast cancer of intermediate risk of recurrence for whom the Prosigna® testing should provide further information about prognostic risk stratification. Only n=3 cases out of the n=142 initial cases were TNBC using local IHC, but those cases were excluded from further analysis because of not addressing the research objective (correlation of IHC-/grade-based vs. molecular subtyping in HR+/HER2- cases). Interestingly, those n=3 TNBC cases were of Prosigna® Luminal A subtype and in the low and intermediate Prosigna® risk group, respectively. However, due to this small TNBC sample size, meaningful statistical statements cannot be made.

2) We highly acknowledge this comment. As we have stated in the limitations [page 11, lines 357-366: ”However, we were not able to track any changes, either in gynecological oncologists-recommended therapy or patient’s attitude, in favor of adding adjuvant chemotherapy. To address this issue, an ongoing single-center study (C1) is currently evaluating prospective questionnaires to assess patients’ decision about adding adjuvant chemotherapy before and after molecular testing. Furthermore, due to missing survival information and small number of cases, we could not correlate Prosigna® risk estimation with patients’ outcome and menopausal status. Therefore, we could analyze which surrogate subtyping matches best with Prosigna® subtyping but not which subtyping method (IHC vs. IHC+G, local vs. central, surrogate vs. molecular subtyping) has the greatest impact on prognosis.”], we are not able to answer this issue for the cohort described in the manuscript. However, as stated above, we currently perform a single-center study addressing this question.

Moreover, being very aware of the clinical relevance and importance of correlation of our results with clinical data, we have to admit that, unfortunately, we do not have access to most of the survival data of our cohort. The lack of clinical parameters is mainly attributable to those cases for which routine diagnostic was performed in laboratories not offering Prosigna® testing assay and that turned to central institutes (i.e. C1-C5) for gene expression testing only. I.e., most patients were diagnosed and treated at smaller decentral sites with the pathological laboratories not being able to perform the Prosigna® test by themselves. If the clinicians of this hospitals wished for Prosigna® testing, a suitable FFPE block plus information on tumor size and nodal status was delivered to the study sites C1-C5 but not any further clinical data. In addition, although Prosigna® examination was sometimes ordered as an inpatient procedure, further treatment/follow-up was provided by outpatient oncologists who were not known to the decentralized pathologists. Therefore, it was almost impossible to obtain more clinical information about the patients in our cohort (briefly mentioned on page 11, lines 350-353).

Best regards.

Round 2

Reviewer 2 Report

No more comments.